# Comparison of E-Textile Techniques and Materials for 3D Gesture Sensor with Boosted Electrode Design

**DOI:** 10.3390/s20082369

**Published:** 2020-04-22

**Authors:** Josue Ferri, Raúl Llinares Llopis, Gabriel Martinez, José Vicente Lidon Roger, Eduardo Garcia-Breijo

**Affiliations:** 1Textile Research Institute (AITEX)-Alicante, 03801 Alcoy, Spain; josue.ferri@aitex.es (J.F.); gmartinez@aitex.es (G.M.); 2Departamento de Comunicaciones, Universitat Politècnica de València, 03801 Alcoy, Spain; rllinares@dcom.upv.es; 3Instituto Interuniversitario de Investigación de Reconocimiento Molecular y Desarrollo Tecnológico (IDM), Universitat Politècnica de València, Universitat de València-Valencia, 46022 Valencia, Spain; jvlidon@eln.upv.es

**Keywords:** gesture recognition, e-textile techniques, e-field sensors, wearables, touchless

## Abstract

There is an interest in new wearable solutions that can be directly worn on the curved human body or integrated into daily objects. Textiles offer properties that are suitable to be used as holders for electronics or sensors components. Many sensing technologies have been explored considering textiles substrates in combination with conductive materials in the last years. In this work, a novel solution of a gesture recognition touchless sensor is implemented with satisfactory results. Moreover, three manufacturing techniques have been considered as alternatives: screen-printing with conductive ink, embroidery with conductive thread and thermosealing with conductive fabric. The main critical parameters have been analyzed for each prototype including the sensitivity of the sensor, which is an important and specific parameter of this type of sensor. In addition, user validation has been performed, testing several gestures with different subjects. During the tests carried out, flick gestures obtained detection rates from 79% to 89% on average. Finally, in order to evaluate the stability and strength of the solutions, some tests have been performed to assess environmental variations and washability deteriorations. The obtained results are satisfactory regarding temperature and humidity variations. The washability tests revealed that, except for the screen-printing prototype, the sensors can be washed with minimum degradation.

## 1. Introduction

Textiles are everywhere; used as decoration, worn as protection, assembled as interior components in cars, or structural elements in constructions among others. Although normally, the textile industry is considered a traditional sector, the new market demands innovations to distinguish it from the other competitors and meet the new requests of the market [1]. The evolution of electronics and textile technologies have enabled the combination of their strengths, increasing computing speed, reducing chip size, improving energy autonomies and offering flexible surfaces. There exists an interest in new wearable solutions that can be directly worn on the curved human body or integrated into daily objects. Textiles offer some features appropriate to the human body, such as softness, comfortability and flexibility and others that make them suitable to include electronics, since they can be stretched, compressed, twisted and deformed arbitrarily. In addition, they offer excellent integrity and structural behavior during daily wearing and washing. 

One of the most challenging applications of electronic textiles is to analyze postures and gestures for different purposes. The corresponding posture and gesture sensors can be utilized for many applications, such as rehabilitation [2,3,4], sign language recognition [5,6], remote control [7] or medical applications [8] among others.

Many researchers have developed sensing technologies to meet the need of soft wearable devices. Some of them are focused on soft bending sensors based on specific materials to accommodate the flexible and deformable nature of the human body [9]. Elastic conductive webbing solutions [10] or highly flexible fabrics [11] have drawn some interest. These solutions are normally used as small pieces of material attached to clothes in a specific position. Other groups of solutions are embedded in the fabric. These sensors combine different layers of both conductive and non-conductive textiles such as smart sleeves. These smart sleeves with deformable textiles enable the detection of a wide range of deformable gestures [12]. Other approaches present microfiber sensors by means of using fibers, such as dual-core conductive fibers working as a capacitive strain sensor, which can be sewed in a bandage or gloves [13], or more complex materials such as piezoelectric [14,15] or triboelectric [16,17] materials to create fiber strain sensors that can detect bending and torsion deformations. There are also solutions based on stretchable conductive threads that measure variations of resistance using materials such as reduced graphene oxide [18] or PEDOT [19] among others. All these previous solutions need to be attached to clothes or gloves to locate the sensors in specific positions to detect bending movements that sometimes could be obtrusive. As alternative solutions, some authors place the sensor on other external surfaces, obtaining touchless textile sensors used as an input to environmental control for individuals with upper-extremity mobility impairments [20] or multi-electrode capacitive sensors that detect simple control gestures [21] by means of contactless sensing.

Some existing research in textile sensors focuses on the use of capacitive sensors. Textile capacitors can be made combining conductive materials that are acting as conductive plates separated by dielectrics. The conductive plates can be woven, sewn and embroidered with conductive thread/fabrics, or they can be painted, printed, sputtered or screened with conductive inks or conductive polymers. The dielectrics used are typically foams, fabric spacers or non-conductive polymers [22,23]. Some works have achieved capacitive embroidered interdigitated structure used as moisture sensors [24], or conductive-knit fabric used as strain sensors [25]. Recently, RFID antennas have been implemented using embroidery [26]. Other investigation studies the impact of the human body on a capacitive textile sensor concluding how the movement of the body can affect the capacitance [27].

Regarding capacitive textile sensors for gesture recognition, few references can be found in the literature. In [20], 12 textile conductive plates sewn in a fabric implement a textile touchless capacitive sensor. In our previous work, two capacitive sensors for the purpose of gesture recognition were presented [28]. In addition, the behavior and influence of different e-textile materials in the textile sensor were shown. The electrodes that conformed the structure of the standard sensor were printed on textiles substrates using screen-printing technology. The different smart textiles prototypes presented were compared with a reference sensor. 

In the present paper, an own boosted sensor design on a textile substrate is developed. It is based on the design recommended by Microchip. This boosted design presents fewer conductive layers and better performance than the standard one [29]. Although the boosted design needs higher voltage and more power than the standard one, it is also sensitive to a greater distance between the hand and the surface of the sensor. Three textile manufacturing technologies are used to implement this type of sensor with satisfactory results. Characterization of the sensors using a static artificial hand was performed. Subsequently, validation was carried out with different subjects, measuring the detection rate after several gesture repetitions. The obtained prototypes present some features such as flexibility that makes them suitable to be attached to clothing or textiles surfaces such as armchairs, curtains or automotive upholstery.

## 2. Materials and Methods

The research is divided into two main parts, one corresponding to the design and working principles and the other focusing on the development of the prototypes and the obtained results. Four resulting sensors were made considering the design recommendations of Microchip. The corresponding materials and technologies used were conductive ink screen-printed, conductive thread embroidered, conductive fabric thermosealed and conventional printed circuit board (PCB) using a milling machine. The expected theoretical value of the associated capacities is studied considering the permittivity of the materials to be used as dielectric layers. Furthermore, the real capacitance values obtained for each individual electrode are presented and compared with the theoretical ones. Afterward, the sensitivity of each individual development is measured and compared with the standard versions. Next, a characterization of the sensors and a user validation stage was carried out to assess their functioning. Moreover, an additional study of the response considering variations of temperature and humidity is presented. Finally, a complementary study of washability is given.

### 2.1. Electronic Design

Microchip Technology Inc. developed a new technology for gesture recognition [23]. This feature is carried out by means of some specific devices, named MGC3130 and MGC3030 (Microchip Technology Inc, Chandler, AZ, USA), capable of sensing a series of hand or finger movements. These devices, in combination with some sensing electrodes and a gesture recognition algorithm, compose the GestIC^®^ sensor. This sensor utilizes an electric field for advanced proximity sensing. Usually, this sensor is implemented using a printed circuit board (PCB) technology on a rigid or flexible substrate, normally of polymer materials. The signals provided by the sensor are processed by the MGC3XXX devices. These devices utilize an algorithm to detect the position of the hand with respect to the sensor and the following gestures: approach detection, position tracking in 3D, sensor touch (touch, multitouch, tap and double-tap), flick gestures, circle gestures and air wheel.

The basic sensor design consists of 4 or 5 receiving electrodes (Rx) connected to the MGC3XXX Rx pins, 1 transmitting electrode (Tx) connected to the MCG3XXX Tx pin and an isolation layer between Rx and Tx electrodes. Both electrodes, Rx and Tx, are implemented with a conductive material such as copper, silver, etc. Regarding the isolation layer, it is made of any non-conductive material (FR4, glass, PET, etc.).

The sensor is based on the use of an electric field to sense proximity. The lines of the electric field are distorted when a hand or finger is placed in the free space above the sensor (Figure 1). The registered variation allows the system to detect, track or classify the motion resulting in a gesture visualized in a user interface application. The electrical field is generated by the application of an alternating signal conducted by an electrode that acts as an antenna. This produces an electrical field that propagates three-dimensionally around the surface. This electrode is considered the transmission electrode or Tx electrode. Its geometry is much smaller than the used wavelength, forcing the magnetic component to be practically zero with no wave propagation. Thus, quasi-static electrical near field is found, allowing the system to sense conductive objects, such as the human body. When a person places a finger on the sensor, it intrudes into the electrical field. The field lines are drawn to the finger and shunted to the ground using the intrinsic conductivity of the human body. This produces a distortion of the electrical field that is detected using some reception electrodes named Rx electrodes. Different Rx electrodes at different positions enable one to determine the origin of the perturbation and its value.

Microchip proposes two designs for the GestIC^®^ sensors (Figure 1).

Standard sensor (Tx signal amplitude of 2.85 V). It is used in small or medium-sized devices (between 20 and 140 mm of width or length of the sensor). It is mandatory for devices with a weak connection to ground, that is, with battery. This Standard sensor consists of a top layer where four Rx electrodes are located on each of the cardinal points as well as a central Rx electrode. This layer is separated from the bottom layer, which contains the Tx electrode, by a dielectric. The ground plane layer is optional and would be located below the Tx electrode layer.

Boosted sensor (Tx signal amplitude between 5 and 18 V). It is used in systems with a large sensor size (even more than 200 mm of width or length of the sensor). In this case, the system must necessarily be grounded. This Boosted sensor consists of a top layer where four Rx electrodes are located on each of the cardinal points and a central Tx electrode. This layer is separated from the bottom layer that contains the Ground (GND) plane by a dielectric. The sensitive area is just delimited by the four perimeter Rx sensors.

The design of the boosted sensor is based on the structure of the Standard sensor with some modifications (Figure 2). More information on the working principle and the design of the Standard sensor can be found in the study by Ferri [22]. 

The main difference between the standard and the boosted sensor consists of the location of the Tx electrode. Boosted Tx electrodes are not placed underneath the Rx electrodes. Instead, the Tx electrode is laid out in the same layer as the Rx electrodes, substituting the central Rx electrode. Microchip recommends placing a GND plane in the bottom layer.

The design of the Rx electrodes is identical to the Standard sensor. For a sensor length of <140 mm, the electrode’s width must be between 4% and 7% of its electrode’s length whereas for a sensor length of >140 mm, it is recommended to use a width between 5 and 7 mm. As aforementioned, there is no Rx center electrode, since the center is dedicated to the Tx electrode.

Regarding the Tx electrode, it is not placed on another layer, but in the center of the Rx frame. This change, with respect to the standard design, forces a minimum distance between the Rx and Tx electrodes to limit the noise coupling between them. This distance is about 3 and 5 mm. 

The Rx feeding lines must be kept apart from the Tx electrode. They must be routed next to the Rx electrodes or inside the GND plane. In addition, it is necessary to keep a distance of about 0.3 or 0.5 mm between GND and the Rx feeding lines.

Regarding the number of layers, the Standard sensor can be designed with a 2- or 3-layer stack, usually with 3 layers. On the other hand, the Boosted sensor consists of a 2-layer stack, a top layer with the Rx and Tx electrodes and a bottom layer with the GND plane. The optimum distance between these two layers (t) will depend on the relative permittivity (ε_r_) of the isolation material between both layers. Microchip recommends t > ε_r_/5. Hence, for an FR4 material (glass-reinforced epoxy laminate material with εr = 5), the thickness can be of 1 mm, whereas for plastic (ε_r_ = 3), it can be of 0.6 mm.

With respect to the circuitry for the boosted sensor, it is necessary a Tx level shifter since the Tx output voltage from MGC3XXX is 2.85 V. Microchip recommends the use of an MCP1416 power driver which allows one to increase the amplitude to values of up to 18 V. Hence, an additional DC input voltage must be used, or it must be generated by a DC–DC boost converter such as the MCP1661 converter.

### 2.2. Materials

With the aim of finding a textile with good characteristics to obtain capacitances of the order of 20 pF (capacitance recommended by Microchip), different cotton, polyester and mixed fabrics with different fabric densities, yarn diameter and weave were studied in the study by Ferri [22,30]. According to these studies, a polyurethane fabric with excellent characteristics was found. Table 1 shows the main characteristics of that fabric. 

The conductive ink used was nano-silver DGP–NO from ANP (Table 2) for screen-printing technology, silver-coated polyamide/polyester hybrid thread (Silvertech 120) from AMANN Group (Table 3) for E-broidery and conductive woven fabric plain Shieldex^®^ Zell from STATEX (Table 4) for the direct application of conductive textiles.

### 2.3. Manufacturing

In this research, an own boosted sensor design has been developed to be applied to a textile substrate. It is based on the design recommended by Microchip and applied to different manufacturing technologies, such as screen-printing technology, E-broidery and direct application of conductive textiles.

The screen-printing process consists of forcing pastes of different characteristics over a substrate through some screens using squeegees. Openings in the screen define the pattern that will be printed on the substrate by serigraphy. Conductive silver ink has been employed to make the electrodes and the GND plane.

E-broidery consists of patterning of conductive textiles by numerically controlled sewing or weaving processes. Conductive silver-coated threads have been used to embroider the electrodes and the GND plane.

Direct application of conductive textiles consists of adhering conductive textiles on the textile to create the electrodes and the GND plane.

The capacitances associated with the architecture design are fundamental in the operation of the sensor and, therefore, in its design. The C_TxRx_ (capacitance between the Tx and the Rx electrodes), C_RxGND_ (capacitance between the Rx electrodes and GND) and C_TxGND_ (capacitance between the Tx electrodes and GND) associated capacitances are the ones that must be considered for the design. More information about these capacitances can be found in the study by Ferri [22]. Microchip recommends a value lower than 20 pF for the C_TxRx_ and C_RxGND_ capacitances, and lower than 1 pF for the C_TxGND_ capacitance in the case of Standard sensor, whereas for a Boosted sensor, the values will depend on the booster driving circuit capabilities. 

The proposed design is shown in Figure 3**.** It consists of a ground plane layer (Figure 3a) and a layer containing the Tx electrode, the four Rx electrodes and their connection lines with the MGC3XXX device (Figure 3b). The dimensions are shown in the same Figure. The electrode size is 80 × 80 mm and the sensing area is 66 × 66 mm.

The resulting sensors have been named as 3DBS_Screen, 3DBS_Embroidery and 3DBS_Fabric in the case of those made with the aforementioned e-textile technologies. For comparison purposes, a fourth boosted sensor made with a PCB technology, named 3DBS_PCB, was made. Moreover, two additional Standard sensors have also been analyzed in order to compare the standard and the boosted response. These standard sensors can be found in in the study by Ferri [28]. The names used for them in the paper are 3DS_Screen and 3DS_PCB.

### 2.4. Sensor Development

The same pattern design (Figure 3) has been applied to the three e-textile technologies and, in addition, to a PCB sensor in order to be able to carry out a comparison (Figure 4).

Screen-printing technology (Figure 4a): as previously mentioned, screen-printing technology uses a mesh to transfer ink onto a substrate, openings in the screen defining the pattern that will be printed on the substrate. The rest of the screen is made impermeable to the ink by a blocking stencil. The thickness of the screen, among other parameters, defines the final thickness of the ink. When screen-printing technology is used, it is necessary to manufacture frames with the screen mesh for the design. The screen for the conductors was a 230-mesh polyester material (PET 1500 90/230-48 from Sefar). Afterwards, to transfer the pattern to a screen mesh, a UV film Dirasol 132 from Fujifilm was used. The final screen thickness was 74 μm for the screen for conductors. The patterns were transferred to the screen by using a UV light source unit IC-5000 from BCB. Printing was carried out by using E2XL from EKRA screen-printer with a shore 75° hardness squeegee, 60° squeegee angle, 1 mm snap-off, 3.5 bar force and 100 mm/s. After the deposition of the inks, these were cured in an air oven FED-115 from BINDER at 130° C for 15 min.

E-broidery technology (Figure 4b): An embroidery process whereby an embroidery machine is used to create electrical patterns on textiles. One type of silver-coated polyamide/polyester hybrid threads was used with 28 Tex properties on the textile substrate selected. An embroidery machine was preferred instead of a normal stitching machine due to the fact that it has a better stitch quality and a vast range of design possibilities. The process was carried out by using an embroidery machine from ZSK with an F Head, with a 1.5 mm/stitch configuration. The conductive design has been accomplished with 7350 stitches in all.

Direct application of conductive textiles technology (Figure 4c): The textile conductor Shieldex^®^ Zell was placed on a MACbond 0.5 mil PET double-sided coated adhesive tape from MACTAC. A desktop cutting machine Cameo3 from SILHOUETTE was used to make the pattern cut. Subsequently, the different parts of the electrode were peeled off and adhered to the fabric using the other adhesive side of the film.

PCB technology (Figure 4d): A double-sided PCB was made with a ProtoMat S63 from LPKF. The main characteristics of the PCB are 1.6 mm of thickness, 15 μm of copper thickness and the use of FR4 as the isolation material.

### 2.5. Measurement

The relative permittivity measures were carried out with a BK PRECISION 895 LCR meter. The following measurement accessories were used: BK PRECISION TL89K1 Kelvin Clips Leads and a Yokogawa-Hewlett Packard 16451A Dielectric Test Adaptor. The LCR meter was configured to measure a tension level of 1 V, with an average of 64 samples and a low read rate (Level = 1 V, Avg = 64, Mean Time = Low). The measurement mode was Cp and D (parallel capacitance and loss tangent). The capacitance measurements were taken at three different parts of the fabric with a 5-frequency scan (0.1 kHz, 1 kHz, 10 kHz, 100 kHz and 1000 kHz). The ε_r_ value was obtained directly from the C_p_ value.

The capacitance values were measured with the same equipment but in mode Cp-Rp (parallel capacitance and resistance) and Z-d (impedance and phase). The capacitance measurements were taken at three different parts of the fabric with a 9-frequency scan (0.1 kHz, 0.5 kHz, 1 kHz, 5 kHz, 10 kHz, 50 kHz, 100 kHz, 500 kHz and 100 kHz).

## 3. Results and Discussion

Figure 5 shows the relative permittivity values (εr) of the fabric which acts as a dielectric of the capacitor. This value is important since it defines the values of the capacitances of the electrodes. As can be seen, its value remains stable from 104 Hz. Since the working frequency of MGC3XXX is situated between 40 kHz and 100 kHz, the relative permittivity remains constant and does not contribute to a variation of the capacity value in this working range [30].

The relative permittivity allows one to calculate the expected theoretical value of the associated capacities. The edge effect capacitance value (C_edge_) has been taken as the theoretical value since it considers the effect of the field lines around the edges of the capacitor. It is estimated using Equations (1) and (2):(1)$$C_{edge}=\frac{\varepsilon_{0}\cdot \varepsilon_{r}\cdot \left(L+\Delta f\right)\cdot \left(w+\Delta f\right)}{t}$$
(2)$$\Delta f=t+\frac{\varepsilon_{0}\cdot t\cdot 10\cdot \ln\left(\left(L+w\right)+1\right)}{\pi }$$
where *C_edge_* is the value of the capacitance in pF, *L* is the length in cm, *w* is the width in cm, *t* is the thickness in cm, *ε_r_* is the relative permittivity and *ε_0_* is the vacuum permittivity (8.85 × 10^−12^ F/m). The capacitance value (C_RxGND_) of the North electrode has been calculated since this electrode is the one with less influence of the capacitances associated with the Rx conduction lines to the connector. The calculated value is 10.9 pF at 10^4^ Hz.

For each manufacturing technology and type of textile, the following measures of their electrical parameters have been taken: the C_TxRx_, C_RxGND_ and C_TxGND_ capacitances. Figure 6 shows the Cp and Rp values of the parallel equivalent circuit as well as the impedance and phase values. The frequency response of the capacity value remains constant in the working range of MGC3XXX, ensuring the correct configuration of the device regardless of the frequency value. The Rp value is low enough to not have significant loss current. The impedance and phase correspond to the expected ones in a pure capacitor and there is no resonance up to the studied frequency (10^6^ kHz).

Table 5 shows only the values of the capacitances (Cp) associated with each one of the types of developed sensors. All the boosted sensors have a capacitance lower or close to 20 pF. 3DBS_Fabric presents the lowest capacitances, followed by 3DBS_Screen. 3DBS_Embroidery and 3DBS_PCB have capacitances around 20 pF and a C_TxGND_ noticeably greater than 3DBS_Fabric and 3DBS_Screen; this difference may be what leads to a different sensitivity of these sensors can be seen later. In the case of the standard versions, 3DS_Screen presents a capacitance lower than 20 pF, except in the case of C_TxGND_. For this reason, an operational amplifier (op-amp) must be inserted between the Tx pin and the Tx electrode as Microchip recommends. Finally, 3DS_PCB presents the greatest capacitances, above 20 pF.

When comparing the theoretical C_RxGND_North_ value (10.9 pF) with the real one (C_RxGND_Screen_ = 12.5 ± 0.9 pF, C_RxGND_Embroidery_ = 22.7 ± 1.1 pF and C_RxGND_Fabric_ = 9.9 ± 0.1 pF), it can be observed that the theoretical approximation is very reliable. It is practically identical in the case of using fabric, very approximate in the case of using ink and it moves away significantly in the case of using embroidery. In the latter case, it can be considered that there is an added effect in the dielectric caused by the air that exists between the fabric and the threads and among the threads themselves.

The sensitivity was measured with software provided by Microchip, the AUREA graphical user interface. The sensitivity allows one to characterize the sensors, being, maybe, the most important parameter of these sensors. Microchip [31] provides an “artificial hand”, a 40 × 40 × 70 mm Styrofoam (ε_r_ ≈ 1) cube covered with an adhesive copper sheet (Figure 7). This block must be connected to the ground to simulate the conditions of the human body. To determine the sensitivity at different heights from the surface of the sensor, the block is placed on blocks of Styrofoam (with no copper cover) of different thicknesses (1, 2, …, cm). AUREA allows one to read the obtained data.

Figure 8 shows a representation of the sensitivity in terms of the signal deviation. Microchip references the signal deviation obtained in MGC3XXX with respect to the distance of the hand compared with the width of the Rx electrodes. The 3DBS_Screen and 3DBS_Fabric sensors present the best sensitivity with similar capacitances to the 3DBS_Embroidery and 3DBS_PCB sensors. The difference may be due to the fact that in the latter, the C_TxGND_ is greater. The standard version, 3DS_Screen, presents a similar sensitivity to the 3DBS_Embroidery and 3DBS_PCB sensors. However, the 3DS_Screen design manufacturing needs one more layer, which is a disadvantage, despite having similar capacitances. Another factor to consider is that the boosted sensor uses a Tx output voltage increasing the amplitude up to 18 V. As expected, the PCB standard version presents the lowest sensitivity.

As aforementioned, the sensor is initially designed to be attached to clothing or textiles surfaces such as armchairs, curtains or automotive upholstery. In this case, the most important textile feature is flexibility. Several bend radiuses were proved and the response was successful up to radiuses lower than 3 cm. The driver had to be calibrated for every test. Figure 9 shows the flexibility of the resulting 3DBS_Embroidery sensor, assessing the flexibility feature of the e-textile sensor. 

A study of the response of the electrode based on the variation of temperature and humidity has been carried out to determine their influence under different operating conditions with a climatic test chamber CTS C70/300 from Controltecnia-CTS. Figure 10 shows how these variations have a low influence on the associated capacitance.

Finally, a study of the variation in conductivity versus washing processes has been carried out. The characteristics of the wash have been one hour of delicate wash at 30 °C with a spin cycle of 700 rpm. Figure 11 shows the result of the normalized resistance variation, from initial resistance value, in a conductor pattern used in the three technologies (fabric, thread and ink). 

The best behavior to washing is offered by the conductive thread, maintaining its characteristics after 100 washes. The silver ink deteriorates rapidly after 5 washes, losing conductivity, (Figure 12) and the conductive fabric supports 100 washes but its electrical characteristics varied during the process.

For the silver ink case, it could have chosen to cover the sensor with a heat-sealable film, but after the washing test with this type of film, problems of adhesion between the film and the conductors also arise after few washes (Figure 13). With this background, the best option is to use a conductive thread.

For the characterization of the sensors, the following protocol has been established. An artificial hand has been placed at different positions on the *X*-, *Y*- and *Z*-axis with respect to the center of the sensor. For each position, the signal deviation data provided by the device has been registered. For the sake of simplicity, only the measurements taken from the PCB sensor and the screen-printed sensor (3DBS_Screen) are shown in Figure 14. It shows the curves obtained by moving the artificial hand in the *X*-axis in the [–5 cm, 5 cm] interval in 1 cm steps for the following values of *Z*: 1 cm, 3 cm and 5 cm. In all cases, the value of *Y* = 0.

As can be seen, the results for the textile sensor are not as smooth as in the PCB sensor due to the textile structure itself. On the other hand, a conductivity variation is observed in the textile sensor, as opposed to the case of the PCB sensor. This behavior is due to the asymmetry of the east and the west sensor. The design of the east sensor tracks is shifted to the left favoring a shorter track for the west sensor. The different tests carried out showed the same behavior. Despite this, the functioning of the gesture recognition is not affected, since the variations are high enough to be detected in the five sensors. The rest of the sensors showed similar behavior.

Next, a validation of the sensors was carried out with different subjects. The objective was to measure the detection rate of several gestures. Figure 15 shows two of the gestures used in the validation procedure. The software used, AUREA, was supplied by Microchip.

In the validation procedure, 10 subjects participated in the experiment. Initially, they were completely unaware of the operation of the sensor. They were allowed to interact with the sensor for 10 minutes. Afterwards, they were asked to perform different gestures detectable by the sensor. After calibration, the subjects were asked to make 10 repetitions of the following movement types: flick from west to east, flick from east to west, flick from north to south, flick from south to north, clockwise circular and counterclockwise circular. The results of the detection rate for the 3DBS_Screen sensor are shown in Table 6.

The results show that most of the gestures are detected successfully. All the flick-type gestures were detected with a percentage higher than 75%. In contrast, the clockwise and counterclockwise gestures have a lower detection rate. This is due to the higher complexity of the circular movement. Besides, the counterclockwise is more difficult to detect than the clockwise movement.

It can be observed (Table 7) that although the three proposed sensors have slightly worse performance than the PCB sensor, most of the gestures were detected successfully. The proposed sensors have better performance with flick gestures rather than with counterclockwise and clockwise gestures. Among the proposed sensors, the 3DBS_Embroidery showed the best performance.

## 4. Conclusions

The aim of this work was to obtain a textile 3D gesture recognition sensor based on the Microchip 3D GestIC^®^ sensor design. A boosted sensor was used allowing greater sensing distances with respect to the standard sensor. Three manufacturing techniques were considered as alternatives: screen-printing with conductive ink, embroidery with conductive thread and thermosealing with conductive fabric. The theoretical capacitance study was very reliable when compared with the real measurements. The prototypes were validated with respect to the detection of gestures with successful results. In the base of the studies of sensitivity, response to humidity and temperature, and resistance to washing, it can be concluded that the technique of embroidery with conductive thread is a good solution to incorporate into the e-textile industry.

## Figures and Tables

**Figure 1 sensors-20-02369-f001:**
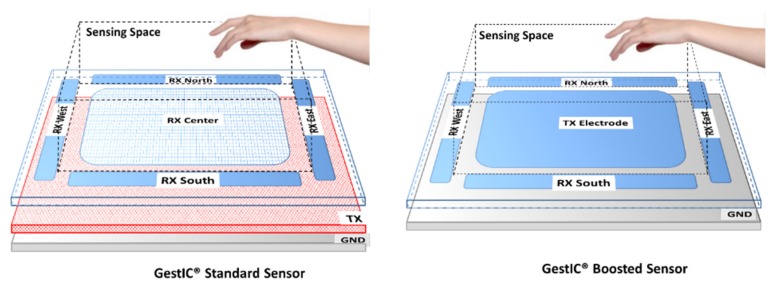
Standard (**left**) and Boosted (**right**) sensors used by Microchip. The standard version consists of 5 RX electrodes on the top layer plus a transmitting electrode (Tx) on an inner layer. The boosted version has 4 receiving electrodes (Rx) and 1 Tx electrodes on the top layer. The sensing space is the same in terms of area but is lower in volume in the case of the standard version. Source: Microchip Technology Inc.

**Figure 2 sensors-20-02369-f002:**
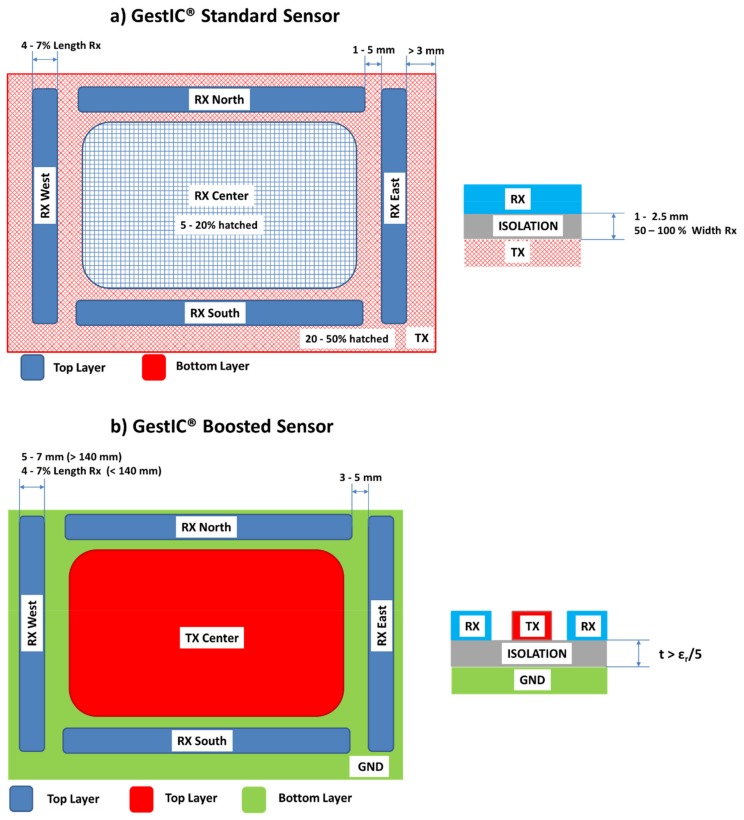
Basic design parameters recommended by Microchip for a (**a**) Standard sensor and (**b**) Boosted sensor. Source: Microchip Technology Inc.

**Figure 3 sensors-20-02369-f003:**
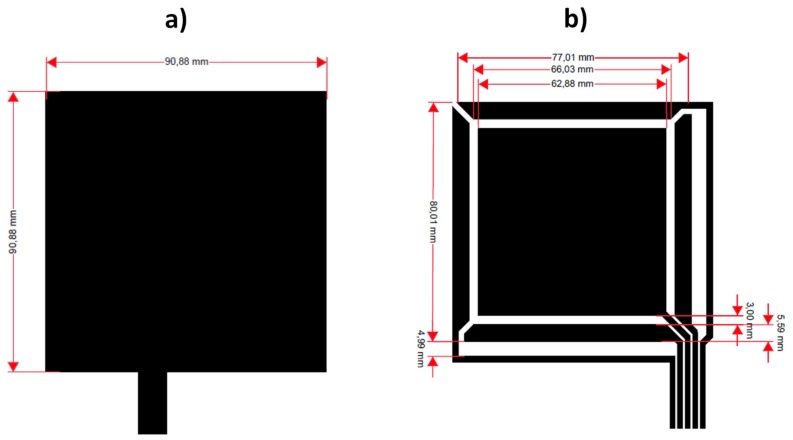
Boosted sensor design: ground plane layer (**a**) and Tx–Rx electrodes (**b**).

**Figure 4 sensors-20-02369-f004:**
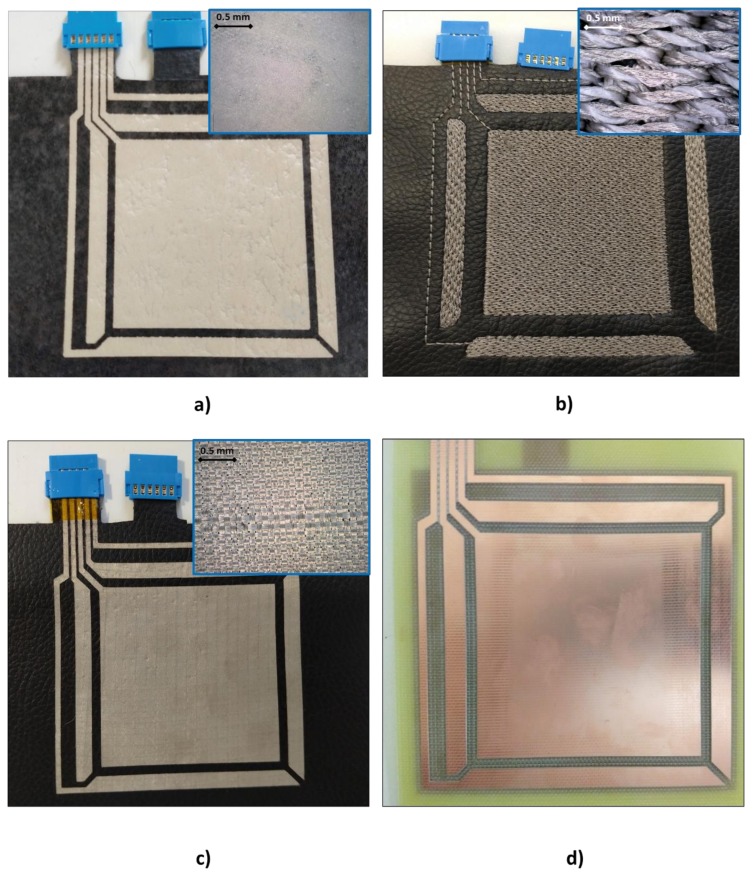
Three-dimensional (3D) gesture sensor developments with the three e-textile technologies: (**a**) screen-printing, (**b**) embroidery and (**c**) direct application of conductive textile and with the printed circuit board (PCB) material (**d**).

**Figure 5 sensors-20-02369-f005:**
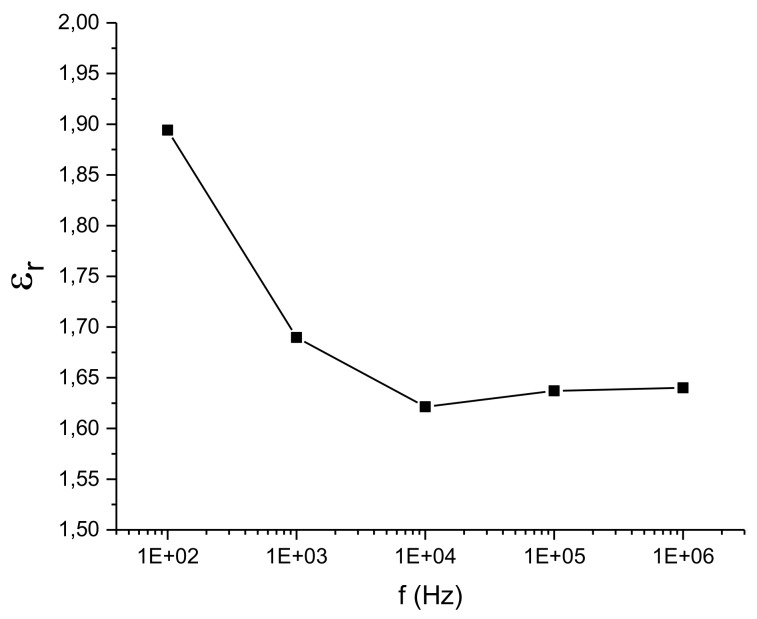
Frequency response of fabric relative permittivity.

**Figure 6 sensors-20-02369-f006:**
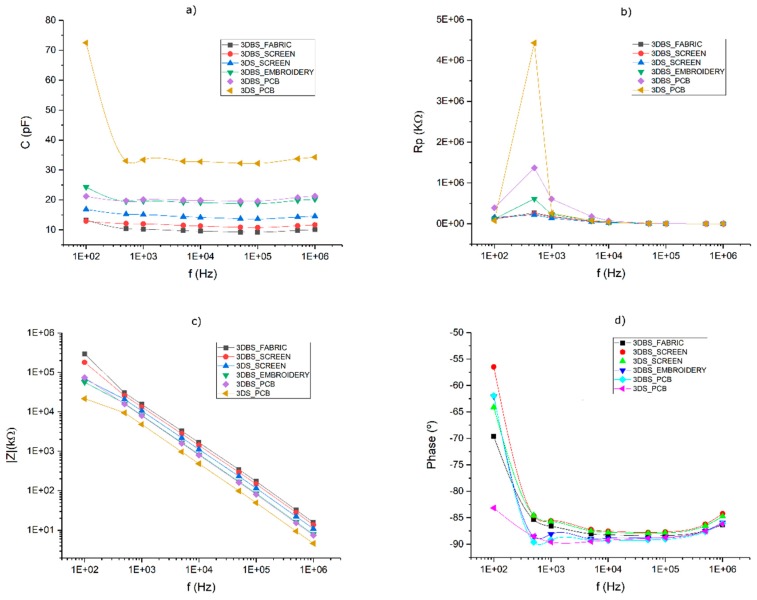
Values of the equivalent capacitor circuit: (**a**) parallel capacitance Cp, (**b**) parallel resistance Rp, (**c**) impedance and (**d**) phase value.

**Figure 7 sensors-20-02369-f007:**
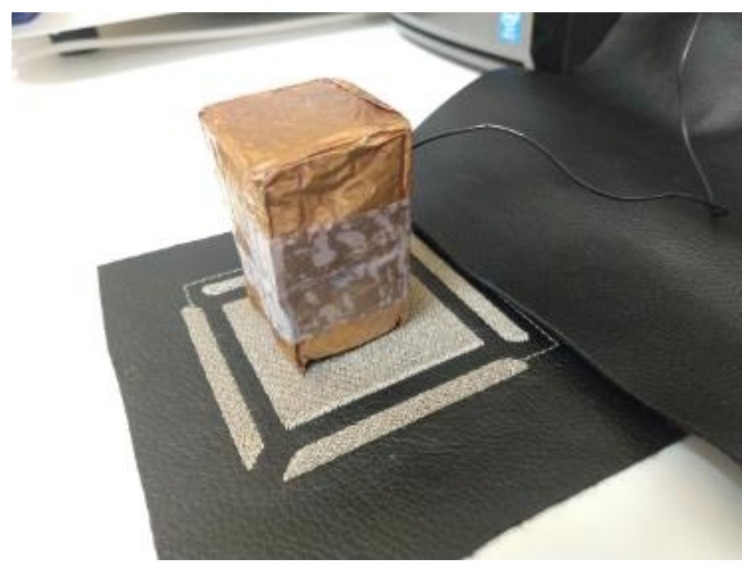
“Artificial hand” provided by Microchip made of Styrofoam covered by copper and connected to ground.

**Figure 8 sensors-20-02369-f008:**
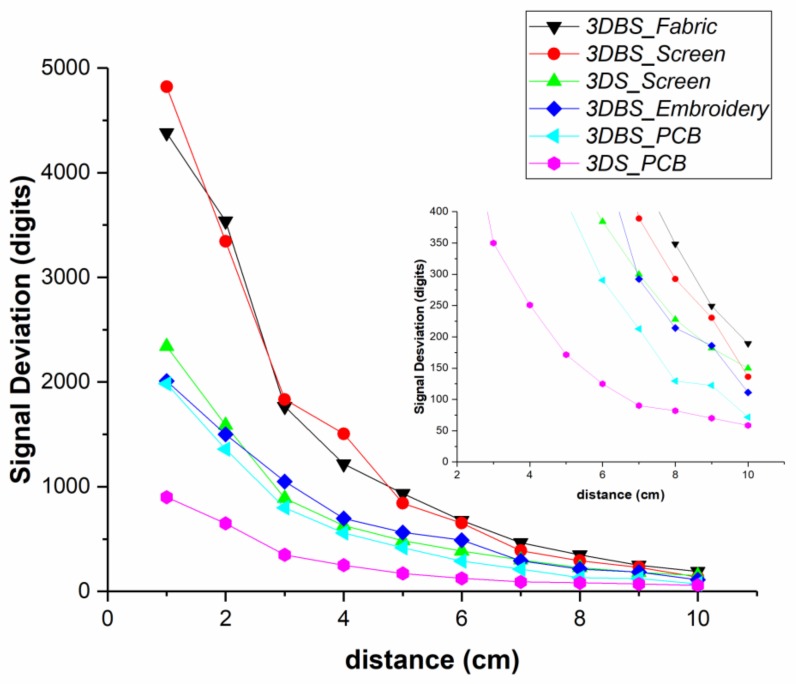
Signal deviation of the different sensors in function of the distance of the hand from the surface of the sensor. The smallest graph is the magnification of the 10 cm limit.

**Figure 9 sensors-20-02369-f009:**
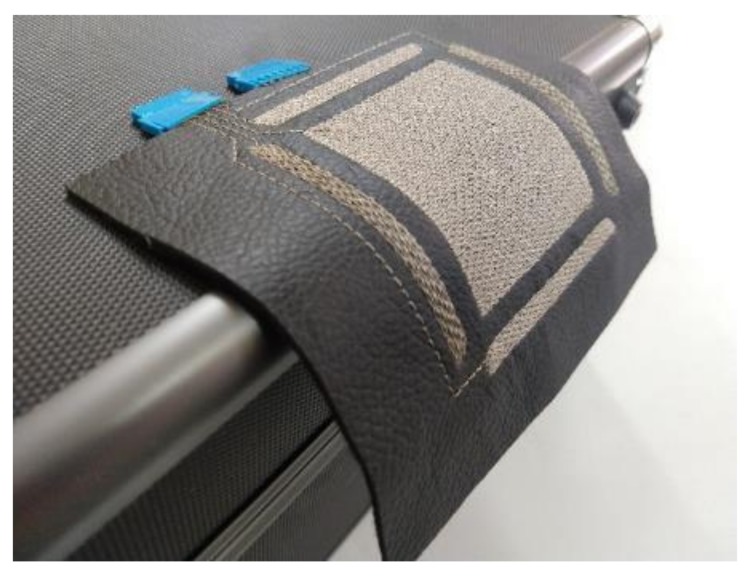
Flexible 3D Boosted sensor manufactured.

**Figure 10 sensors-20-02369-f010:**
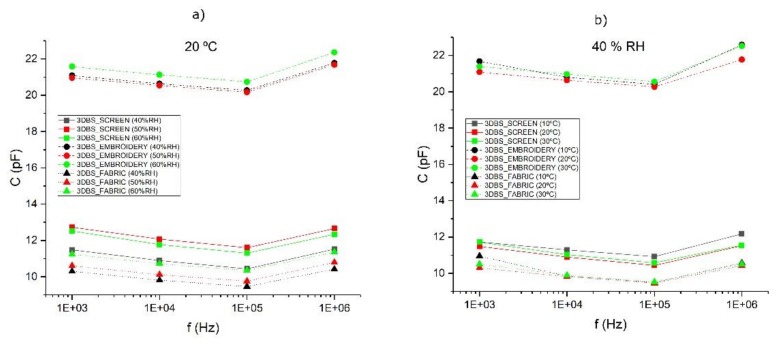
(**a**) Capacitance variation depending on the humidity at a fixed temperature of 20 °C; (**b**) capacitance variation depending on the temperature at a fixed humidity of 40% RH.

**Figure 11 sensors-20-02369-f011:**
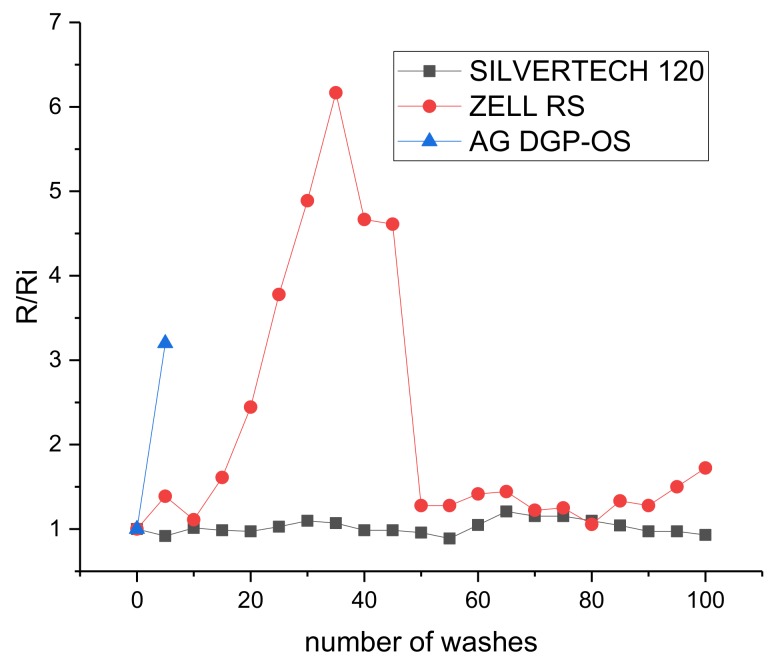
Normalized resistance variation in a conductor pattern used in the three technologies (fabric, thread and ink) versus number of washes.

**Figure 12 sensors-20-02369-f012:**
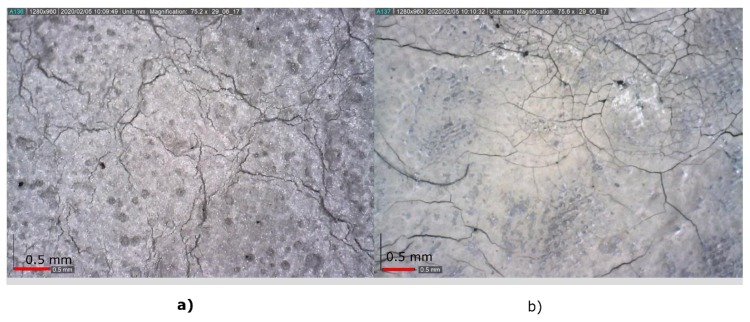
(**a**) Appearance of the silver ink before washing and (**b**) after 5 washes.

**Figure 13 sensors-20-02369-f013:**
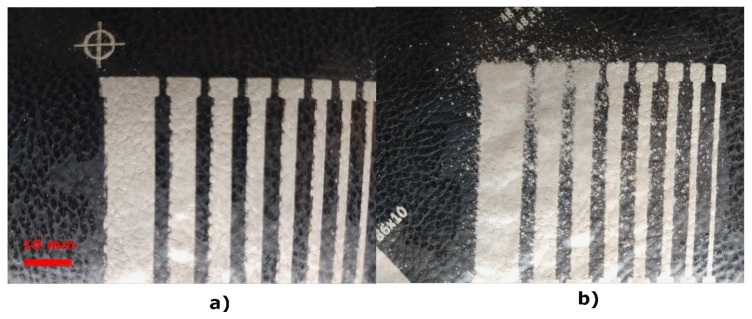
(**a**) Appearance of the silver ink coated with a heat-sealed film before washing and (**b**) after 5 washes. The bubbles generated between the substrate and the film and the degeneration of the silver ink are observed.

**Figure 14 sensors-20-02369-f014:**
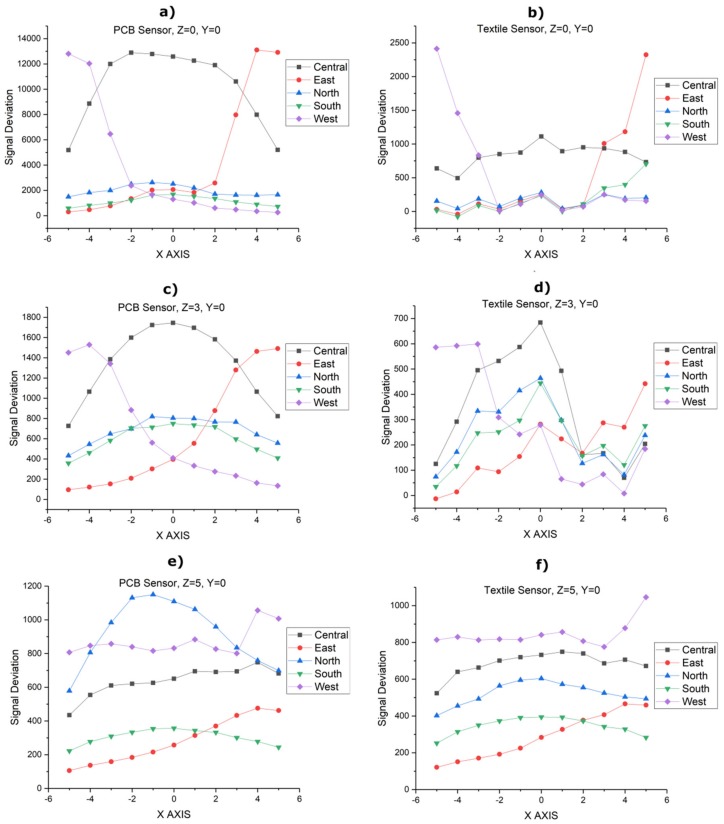
Characterization of the sensors using an artificial hand: (**a**) PCB reference sensor fixing *Z* = 0 position. (**b**) Textile sensor fixing *Z* = 0 position. (**c**) PCB reference sensor fixing *Z* = 3 cm position. (**d**) Textile sensor fixing *Z* = 3 cm position. (**e**) PCB reference sensor fixing *Z* = 5 cm position. (**f**) Textile sensor fixing *Z* = 5 cm position.

**Figure 15 sensors-20-02369-f015:**
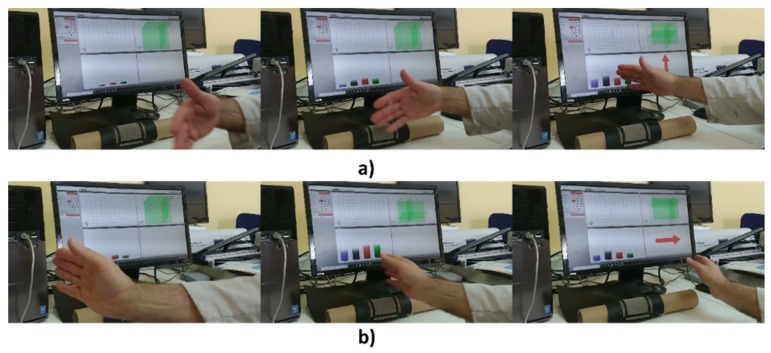
Two of the gestures used in the validation: (**a**) flick from north to south and (**b**) flick from west to east.

**Table 1 sensors-20-02369-t001:** Fabric characteristics.

Fabric	Polyurethane
**Picture**	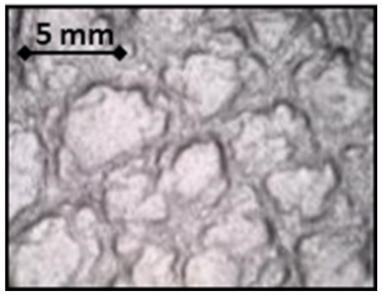
**Weft Material**	Polyurethane
**Warp Material**	Polyurethane
**Ligament**	Non-woven
**Thickness (µm)**	1300 ± 16
**Grammage (g/m^2^)**	152 ± 5

**Table 2 sensors-20-02369-t002:** Silver ink characteristics.

	DGP-NO
**Specific Resistivity (µΩ⋅cm)**	10–50
**Solids (%)**	70–80
**Viscosity (cps)**	50–150
**Curing**	120 °C–150 °C
**Properties**	Silver Nanoparticles

**Table 3 sensors-20-02369-t003:** Embroidery thread silver characteristics.

	Silvertech 120
**Resistance per Unit Length (Ω/m)**	<530
**Textile No.**	28
**Needle Size (in No.)**	11–14
**Materials**	Polyamide/polyester covered of silver
**Properties**	Skin friendly Biocompatible Antibacterial Antistatic

**Table 4 sensors-20-02369-t004:** Conductive fabric plain characteristics.

	Zell RS
**Grammage (g/m^2^)**	77% ± 15%
**Thickness (µm)**	110% ± 15%
**Width (mm)**	1300% ± 3%
**Fabric Density (Thread/cm^2^)**	84 ± 2
**Sheet Resistivity (Ω/sq)**	≤0.02
**Materials**	Tin copper silver plated nylon fabric
**Properties**	Washable

**Table 5 sensors-20-02369-t005:** Capacitance values (pF) at 10^4^ Hz.

	3DBS_Screen	3DBS_Embroidery	3DBS_Fabric	3DBS_PCB	3DS_Screen	3DS_PCB
**C_TxRxN_**	11.7 ± 0.1	18.6 ± 0.5	8.9 ± 0.4	18.5 ± 0.4	15.1 ± 10.3	33.9 ± 10.7
**C_TxRxS_**	13.5 ± 1.0	18.1 ± 0.1	9.8 ± 0.3	24.1 ± 0.4	19.3 ± 10.4	34.6 ± 10.7
**C_TxRxE_**	13.5 ± 0.6	19.8 ± 0.1	11.6 ± 0.6	22.3 ± 0.8	15.1 ± 10.3	30.1 ± 10.6
**C_TxRxW_**	11.6 ± 0.8	16.6 ± 0.5	8.2 ± 0.1	18.9 ± 0.3	16.1 ± 10.3	30.6 ± 10.6
**C_RxNGND_**	12.5 ± 0.9	22.7 ± 1.1	9.9 ± 0.1	20.7 ± 1.1	15.1 ± 10.3	34.3 ± 10.7
**C_RxSGND_**	16.6 ± 0.8	21.8 ± 0.9	11.3 ± 0.2	29.3 ± 1.0	19.5 ± 10.4	33.2 ± 10.7
**C_RxEGND_**	16.1 ± 0,.5	24.1 ± 1.2	12.5 ± 0.2	27.7 ± 0.9	15.1 ± 10.3	30.3 ± 10.6
**C_RxWGND_**	13.0 ± 0.4	19.5 ± 0.7	10.2 ± 0.1	21.4 ± 1.2	16.1 ± 10.3	30.2 ± 10.6
**C_TxGND_**	56.2 ± 0.5	112.6 ± 2.3	38.1 ± 1.1	130.8. ± 5.7	2327.0 ± 56.5	635.0 ± 22.7

**Table 6 sensors-20-02369-t006:** Gesture conducted test results for the 3DBS_Screen sensor.

Gesture	S. 0	S. 1	S. 2	S. 3	S. 4	S. 5	S. 6	S. 7	S. 8	S. 9	%
**Flick W–E**	90%	90%	80%	80%	90%	80%	70%	90%	90%	80%	84%
**Flick S–N**	80%	100%	80%	100%	60%	80%	70%	80%	70%	70%	79%
**Flick N–S**	90%	100%	90%	80%	80%	90%	90%	80%	70%	60%	83%
**Flick E–W**	90%	90%	90%	90%	90%	90%	90%	90%	70%	80%	87%
**Counterclockwise**	60%	70%	70%	70%	70%	60%	60%	70%	60%	60%	65%
**Clockwise**	70%	80%	70%	80%	80%	70%	70%	60%	70%	80%	73%

**Table 7 sensors-20-02369-t007:** Comparison of the results obtained by the three proposed sensors and the PCB sensor.

Gesture	3DBS_Fabric	3DBS_Screen	3DBS_Embroidery	PCB
**Flick W–E**	84%	84.00%	86%	89%
**Flick S–N**	81%	79.00%	81%	81%
**Flick N–S**	84%	83.00%	85%	86%
**Flick E–W**	88%	87.00%	89%	90%
**Counterclockwise**	59%	65.00%	73%	86%
**Clockwise**	76%	73.00%	74%	84%

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
