# Peer review of "Comparison of E-Textile Techniques and Materials for 3D Gesture Sensor with Boosted Electrode Design"

_sensors, 2020, doi:10.3390/s20082369_

Round 1

Reviewer 1 Report

for comments see attached file

Note:There are four pages and there is a little blank in page 3. Please continue to read it until page four.

Author Response

Response to Reviewer 1

Manuscript ID: sensors-751628

The authors would like to thank the Reviewer for the deep and thorough review of this manuscript. The research paper has been revised in the light of the Reviewer’s useful suggestions and comments, hoping to have improved its quality to a better scientific level. Answers to specific comments of the Reviewer are listed below.

  • Comments and Suggestions for Authors: The paper reports about the wearable textile sensor based on the Microchip 3D. Three manufacturing techniques were presented: screen printing with conductive ink, embroidery with conductive thread and thermoseal with conductive fabric. Properties of these three techniques were also compared with the printed circuit board material.I have the following recommendations, questions, and comments:l.

  • Comment #1: Figure 1. and Figure 2. It seems to me to provide very similar information. If these Figs are redundant I recommend presenting only Fig. 2. I also recommend Fig. 2 marking Fig. 2a and Fig. 2b and also provide the caption for this Fig. 2. Figure 2 Basic design parameters recommended a) by Microchip for Standard sensor and b) Boosted sensor. Source: Microchip Technology Inc. if authors think that it is necessary to present both Figs I recommend provide a description of these Figs. in the text in more detail.

Response: We think that it is better to leave both Figures since the first one allows one to observe the sensor’s work area. Anyway, we have added a reference to this Figure at beginning of the paragraph and modified the caption of the Figure. Figure 2 has been modified according to your comments.

  • Comment #2: In Tab.1 on a picture, I recommend inserting the scale.

Response: The scale has been inserted.

  • Comment #3: I would like to ask you for the definition Tex no. mention in Tab. 3?

Response: Tex is the mass in grams of 1,000 meters of thread. If 1,000 meters weigh 25 grams, it is a tex 25. Larger tex numbers correspond to heavier threads [wiki]

  • Comment #4: I recommend adding scales in Fig. 4a, b and c for the zoom pictures.

Response: The scales have been inserted.

  • Comment #5: Row 254: Yokogama-Hewlett Packard 16451A - I expected that authors used Yokogawa- Hewlett Packard instrument.

Response: We have modified the name of the brand and model in the corresponding paragraph.

  • Comment #6: I recommend presented results in Fig. 6 and 7 discuss in more detail and provide a comparison in previously published data. I recommend presenting Figure 7 as 7a, b, c, and d.

Response: We have added a paragraph about relative permittivity. In this case, in Materials section we talked about previous published data.

Figure 7 has been modified according to your comment and a paragraph has been added as well, extending the comments about the results showed in this Figure.

  • Comment #7: I recommend uniting Figs 10 and 11 in Fig. 10 a), b). Figure 10. a) Capacitance variation depending on the humidity at a fixed temperature of 20ºC, b) Capacitance variation depending on the temperature at a fixed humidity of 40% RH.

Response: The Figures have been joined according to the comment.

  • Comment #8: I recommend provide scales for Figures 13. and 14. In Fig. 13 scale is very difficult to read and in Fig. 14 scales are missing.

Response: The scales have been modified and inserted.

  • Comment #9: Minor suggestions.

Response: Thanks very much indeed for the feedback. All minor suggestions have been accepted and changes have been made.

Yours sincerely,

Eduardo García-Breijo

Professor

Universitat Politècnica de València, Spain

Reviewer 2 Report

This article reports the fabric-based flexible gesture recognition sensors. But some issues should be addressed before publication: 1. The authors introduce the designs of gesture recognition sensors in detail, and shows the sensitivity comparison between the standard and boosted sensors. However, they miss the reasons for the higher sensitivity of boosted sensors. They should convincingly explain why boosted sensors have greater sensing distances with respect to the standard sensors in this article. 2. The authors should also consider a convincing experiment to reflect the sensitivity changes of their sensors under different bending conditions. 3. weather the power consumption of the boosted sensors will greatly increase, since the boosted sensors require higher voltage than the standard sensors? 4. The influence on the capacitances of the sensors under different temperature and humidity have been studied in two experiments. But the experimental results cannot be well supported, more experiment under different temperature and humidity conditions should be added.

Author Response

Response to Reviewer 2

Manuscript ID: sensors-751628

The authors would like to thank the Reviewer for the deep and thorough review of this manuscript. The research paper has been revised in the light of the Reviewer’s useful suggestions and comments, hoping to have improved its quality to a better scientific level. Answers to specific comments of the Reviewer are listed below.

  • Comments and Suggestions for Authors: This article reports the fabric-based flexible gesture recognition sensors. But some issues should be addressed before publication:

  • Comment #1: The authors introduce the designs of gesture recognition sensors in detail, and shows the sensitivity comparison between the standard and boosted sensors. However, they miss the reasons for the higher sensitivity of boosted sensors. They should convincingly explain why boosted sensors have greater sensing distances with respect to the standard sensors in this article.

Response: The sensitivity of a standard 3D is half that of the 3D Booster (Figure 9, 3DBS_Screen vs. 3DS_Screen). For the screen-printing type, the difference is 2400 digits in the Standard versus 5000 in the Boosted. The article is focused on finding a cheaper alternative in terms of manufacturing compared to the standard 3D design. Boosted removes a print layer. On the contrary, higher voltages are necessary, we go from 2.85 to 18V, as explained in the article. In addition to reducing a layer, greater distances are achieved, everything influences but for the most part it is due to the increase in voltage.  

  • Comment #2: The authors should also consider a convincing experiment to reflect the sensitivity changes of their sensors under different bending conditions.

Response: The sensitivity is very similar for different flexion angles, so this difference has not been reflected.

  • Comment #3: weather the power consumption of the boosted sensors will greatly increase, since the boosted sensors require higher voltage than the standard sensors?.

Response: Indeed, the boosted has higher consumption, however it is perfectly usable with battery. We have not carried out a study of power consumption, but we have tested two portable prototypes with bluetooh for the Standard and for the Boosted versions and we have not noticed a significant difference in the duration of the battery. Despite this, it would be interesting to study the power consumption.

  • Comment #4: The influence on the capacitances of the sensors under different temperature and humidity have been studied in two experiments. But the experimental results cannot be well supported, more experiment under different temperature and humidity conditions should be added.

Response: We agree that the experiments on temperature and humidity could be expanded, but we did not think it was necessary. The obtained results did not predict a large variation in capacitance. In any case, it is impossible to carry out new experiments at this time since the laboratories are closed due to the coronavirus.

Yours sincerely,

Eduardo García-Breijo

Professor

Universitat Politècnica de València, Spain

Reviewer 3 Report

Three manufacturing techniques for textile contact-less sensor were tested: screen printing with conductive ink; embroidery with conductive thread; and thermoseal conductive fabric.  In principle, the sensors are meant for gesture-recognition applications. The scientific justification of the study was not appropriately depicted. The authors provided extensive design details as well as characterization of the sensors. However, the clarity of those details in the manuscript is poor, and the study lacks a real validation of the sensors. This paper represents a very low contribution to the field.

Comments:

  • The paper lacks of a definite scientific justification. The demand of wearable sensing textiles itself cannot be considered a motivation for a scientific study. Nevertheless, such call for such technology must be objectively supported.
  • The manuscript includes no validation of the sensors. It only has a figure (15) that depicts a single case of swipe to the left and swipe to the right in a single subject.
  • Abstract should include more quantitative or more detailed qualitative assessment of the sensors tested in the study. Later I realized there is no such a quantitative evaluation to include in the abstract.
  • If the authors decide to turn the paper into a gesture-recognition study using textile sensors, it needs a thorough edition. Furthermore, the authors should highlight the results and limitations of previous studies, to determine the novelty of their study.  The introduction has a couple of paragraphs on the state of the art, but it is limited to mentioning previous studies, and no analysis was provided.
  • When the gesture-recognition from Microchip Technology is mentioned, it is not clear what kind of technological approach it is (e.g. EMG, video, gyroscopes, etc.). it looks like it is a proximity sensor, but how the proximity is used to define gesture is not clearly stated. The text is not clear in conveying a straightforward explanation of the functioning of the devices.  Besides, if this is not the design of the authors, it shouldn’t be described with such level of detail. If it is partially work of the research group, their specific contribution should be clearly stated.
  • Table 1, 2, 3, are irrelevant. If relevant for the designe, those details can be provided in the text (alternatively in appendix).  Tables must be kept for results of the study. Some figures are also irrelevant (8, 9, 10, 11, 12)
  • The details on the electrical properties of the sensors should be accompanied with the relevance of each parameter, and described with a sufficient still succinct level of detail. The attention should be driven by the accuracy of the gesture recognition provided by each sensor.
  • How were comfort, conformability, flexibility, and lightness assessed? The authors have claimed that the e-textiles provided those features.
  • Extensive revision of the English language is required. (for example, the use of “Sensibility”: Sensitivity is a more technical term; hand recognition is not the same as gesture-recognition)
  •  

Reviewer 4 Report

This paper compares several wearable-textile manufacturing techniques for 3D gesture sensing, that are: 1) screen printing with conductive ink, 2) embroidery with conductive thread, 3) thermoseal with conductive fabric. The authors provide a theoretical capacitance study demonstrating good reliability in comparison with experimental measurements. Based on their study of sensitivity, response to humidity and temperature, and resistance to washing, they conclude that the technique of embroidery with conductive thread is a good solution to incorporate into the e-textiles.

The paper is well written and technically focused. It definitely will be interesting to the readership of this journal.

As a minor note, there are a few points that should be fixed, as reported below:

P.9, L.238: The reference to figure appears to be broken: "Figure 4Figure 4.c";

P.12, L.320-323: Some characters appear erroneously in red colour;

P.13, L332: The statement "test chamber CTS C70/300 from Controltecnia-CTS" needs to be further elucidated by a reference.

Author Response

Response to Reviewer 4

Manuscript ID: sensors-751628

The authors would like to thank the Reviewer for the deep and thorough review of this manuscript. The research paper has been revised in the light of the Reviewer’s useful suggestions and comments, hoping to have improved its quality to a better scientific level. Answers to specific comments of the Reviewer are listed below.

  • Comments and Suggestions for Authors: This paper compares several wearable-textile manufacturing techniques for 3D gesture sensing, that are: 1) screen printing with conductive ink, 2) embroidery with conductive thread, 3) thermoseal with conductive fabric. The authors provide a theoretical capacitance study demonstrating good reliability in comparison with experimental measurements. Based on their study of sensitivity, response to humidity and temperature, and resistance to washing, they conclude that the technique of embroidery with conductive thread is a good solution to incorporate into the e-textiles. The paper is well written and technically focused. It definitely will be interesting to the readership of this journal.

  • Comment #1: 9, L.238: The reference to figure appears to be broken: "Figure 4Figure 4.c".

Response: Thanks for the feedback. The mistakes have been corrected.

  • Comment #2: 12, L.320-323: Some characters appear erroneously in red colour.

Response: Thanks for the feedback. The mistakes have been corrected.

  • Comment #3:13, L332: The statement "test chamber CTS C70/300 from Controltecnia-CTS" needs to be further elucidated by a reference

Response: We are sorry but we do not understand the comment. Is it necessary to clarify what a climate chamber is or the manufacturer or to indicate a bibliographic reference?

Yours sincerely,

Eduardo García-Breijo

Professor

Universitat Politècnica de València, Spain

Reviewer 5 Report

This work presents a hand recognition touchless sensor, based on the “Microchip 3D GestIC sensors” technology. Four sensor manufacturing techniques were compared (PCB, screen printing with conductive ink, embroidery with conductive thread, and thermoseal with conductive fabric); Evaluating Standard and Boosted sensor technologies. Stability and strength of the solutions were evaluated, and additional tests were performed to assess environmental performances.

The paper is well written, and the contribution is relevant for the field.

Author Response

Response to Reviewer 5

Manuscript ID: sensors-751628

The authors would like to thank the Reviewer for the deep and thorough review of this manuscript. The research paper has been revised in the light of the Reviewer’s useful suggestions and comments, hoping to have improved its quality to a better scientific level.

  • Comments and Suggestions for Authors: This work presents a hand recognition touchless sensor, based on the “Microchip 3D GestIC sensors” technology. Four sensor manufacturing techniques were compared (PCB, screen printing with conductive ink, embroidery with conductive thread, and thermoseal with conductive fabric); Evaluating Standard and Boosted sensor technologies. Stability and strength of the solutions were evaluated, and additional tests were performed to assess environmental performances.
  • The paper is well written, and the contribution is relevant for the field..

Response: Thank you very much for your kind comments.

Yours sincerely,

Eduardo García-Breijo

Full Professor

Universitat Politècnica de València, Spain

Round 2

Reviewer 2 Report

The authors have answered all the questions and can be considered for acceptance, but in my opinion, the novelty is still not enough.

Author Response

esponse to Reviewer 2 Round 2

Manuscript ID: sensors-751628

The authors would like to thank the Reviewer for the deep and thorough review of this manuscript. The research paper has been revised in the light of the Reviewer’s useful suggestions and comments, hoping to have improved its quality to a better scientific level.

  • Comment #1: The authors have answered all the questions and can be considered for acceptance, but in my opinion, the novelty is still not enough.

Response:

Thanks for your response. Regarding the novelty of the paper, it is focused on:

  • Implementation of a new 3D touchless design: a boosted design allowing one greater distances to detect gestures

  • Use of different textile technologies to implement the proposed sensors. In the case of ebroidery and thermosealing solutions, it is the first case of capacitive sensor application on them.

We appreciate that you consider the paper for acceptance. Perhaps the degree of novelty is not maximum, but from the point of view of the use of textile materials it has been a challenge to manufacture with textile materials. We have also focused efforts on making this sensor using different manufacturing techniques in order to achieve the greatest possible number of applications. However, we understand that perhaps the novelty of the developments is not fully explained. This is why we have included more details about the differences in the results obtained with respect to those existing in previous research. In addition we have also included validation results obtained in order to reinforce this aspect.

Yours sincerely,

Eduardo García-Breijo

Full Professor

Universitat Politècnica de València, Spain

Reviewer 3 Report

In response to comment 4, the authors have stated “The purpose of the paper is not a gesture-recognition study using textile sensors, but the technological development of this kind of sensors.”  However, the abstract reads “In this work, a novel solution of a hand recognition touchless sensor is implemented with satisfactory results.” The contradiction is remarkable.  The problem is not the abstract, because they argued that “the abstract faithfully reflects the work. None of the other 4 reviewers have made any comment on it.”

The authors opposed to make clear (succinctly) some aspects of the manuscript: Functioning of proximity sensing, the specifics of Microchip technology, the need for or the relevancy of the figures, methods for the assessment of comfort, conformability, flexibility, and lightness (which they claim their sensors meet), their use of the word sensibility (which refers to stimuli “emotional or moral in nature”), etc.

The whole paper is written like a novel sensor is being presented. The abstract reads “Moreover, three manufacturing techniques have been considered as alternatives: screen printing with conductive ink, embroidery with conductive thread and thermoseal with conductive fabric. The main critical parameters have been analyzed for each prototype including the sensibility of the sensor, which is an important and specific parameter of this type of sensor”, but the response of the authors insists with “this kind of sensors have been validated and the results published in previous publications (referenced in the same paper).” They might refer to reference 22, in which there is no validation of gesture recognition either, but of the electrical and mechanical characteristics of the sensors.  There is a big discrepancy between what the authors understand by validation (of a solution of a hand recognition touchless sensor) and my interpretation of the word.  The comment 2 still holds completely.

The authors changed my words to “wearable sensing textiles itself cannot be considered a motivation for a scientific study”. I said “The demand of wearable sensing textiles itself cannot be considered a motivation for a scientific study”.  In other words, one can argue that there is a need for seamless gesture-recognition wearable technology, for physiological signals from wearable devices, among other scientific issues, but the demand of wearable sensing textiles (as it is a too broad term)  is hard to understand as a scientific motivation for a research study.

The manuscript (first sentence of the abstract) starts with a typo “There in an interest”.

Author Response

Response to Reviewer 3 Round 2
Manuscript ID: sensors-751628

The authors would like to thank the Reviewer for the deep and thorough review of this manuscript. The research paper has been revised in the light of the Reviewer’s useful suggestions and comments, hoping to have improved its quality to a better scientific level. 

- Comment #1:  In response to comment 4, the authors have stated “The purpose of the paper is not a gesture-recognition study using textile sensors, but the technological development of this kind of sensors.”  However, the abstract reads “In this work, a novel solution of a hand recognition touchless sensor is implemented with satisfactory results.” The contradiction is remarkable.  The problem is not the abstract, because they argued that “the abstract faithfully reflects the work. None of the other 4 reviewers have made any comment on it.”.

Response:

The purpose of the paper is not a gesture-recognition study, but the implementation of a sensor for gesture recognition. In addition, we give details about its technological development. As we understand, the study of gesture-recognition in textile sensors is a wide topic involving for instance the implementation of new detection algorithms. 

Nevertheless, as you suggest in this review and in the previous review, we have made some major changes in the paper introducing the part corresponding to gesture recognition.

Abstract
We have corrected some English mistakes and included more quantitative or more detailed qualitative assessment of the sensors tested in the study:

“In addition, a user validation has been performed, testing several gestures with different subjects. During the tests carried out, flick gestures obtained detection rates from 79% to 89% on average.”

“The obtained results are satisfactory regarding temperature and humidity variations. The washability tests revealed that, except for the screen-printing prototype, the sensors can be washed with a minimum degradation.”

Introduction
We have added new references with the results and limitations of previous studies to determine the novelty of their study:

“Some existing research in textile sensors focuses on the use of capacitive sensors. Textile capacitors can be made combining conductive materials that are acting as conductive plates separated by dielectrics. The conductive plates can be woven, sewn, and embroidered with conductive thread/fabrics, or they can be painted, printed, sputtered, or screened with conductive inks, or conductive polymers. The dielectrics used are typically foams, fabric spacers or non-conductive polymers [22,23]. Some works have achieved capacitive embroidered interdigitated structure used as moisture sensor [24], or conductive-knit fabric used as strain sensors [25]. Recently RFID antennas has been implemented using embroidery [26]. Other investigation studies the impact of the human body on a capacitive textile sensor concluding how the movement of the body can affect the capacitance [27].

Regarding capacitive textile sensors for gesture recognition, few references can be found in the literature. In [20], 12 textile conductive plates sewn in a fabric implement a textile touchless capacitive sensor. In our previous work, two capacitive sensors for the purpose of gesture recognition were presented [28]. In addition, the behavior and influence of different e-textile materials in the textile sensor were shown. The electrodes that conformed the structure of the standard sensor were printed on textiles substrates using screen printing technology. The different smart textiles prototypes presented were compared with a reference sensor. 

In the present paper, an own boosted sensor design on a textile substrate is developed. It is based on the design recommended by Microchip. This boosted design presents fewer conductive layers and better performance than the standard one [29]. Although the boosted design needs higher voltage and more power than standard one, it is also sensitive to greater distance between the hand and the surface of the sensor. Three textile manufacturing technologies are used to implement this type of sensor with satisfactory results. A characterization of the sensors using a static artificial hand was performed. Subsequently, a validation was carried out with different subjects, measuring the detection rate after several gesture repetitions. The obtained prototypes present some features such as flexibility that makes them suitable to be attached into clothing or textiles surfaces such as armchairs, curtains or automotive upholstery.”

Results and Discussion

We refer to the response to Comment #3.

- Comment #2:  The authors opposed to make clear (succinctly) some aspects of the manuscript: Functioning of proximity sensing, the specifics of Microchip technology, the need for or the relevancy of the figures, methods for the assessment of comfort, conformability, flexibility, and lightness (which they claim their sensors meet), their use of the word sensibility (which refers to stimuli “emotional or moral in nature”), etc..

Response: 

The functioning of proximity sensing is described in the text in the lines 106-119. We have added the following text to describe the gesture recognition procedure:

“The signals provided by the sensor are processed by the controller. The MGC3130 and MGC2120 controller utilizes an algorithm to detect the position of the hand respect the sensor and the following gestures: approach detection, position tracking in 3D, sensor touch (touch, multitouch, tap and double tap), flick gestures, circle gestures, and airwheel.”

The specifics of Microchip technology is described in the text in the lines 120-173 and Figures 1 and 2. This description is similar to the ones used in other papers [1, 2]:

1. Ferri, J., Llopis, R., Moreno, J., Ibañez Civera, J., Garcia-Breijo, E., 2019. A Wearable Textile 3D Gesture Recognition Sensor Based on Screen-Printing Technology. Sensors 19, 5068. https://doi.org/10.3390/s19235068.
2. Ferri, J.; Lidón-Roger, J.; Moreno, J.; Martinez, G.; Garcia-Breijo, E. AWearable Textile 2D Touchpad Sensor Based on Screen-Printing Technology. Materials 2017, 10, 1450

The need for or the relevancy of the figures is the following:
- Figure 7 (formerly Figure 8): it is the illustration of the “artificial hand”. Since it is an abstract concept, we considered interesting to show it. It could be removed
- Figure 8 (formerly Figure 9): it is an important figure since it shows the behavior of the sensors against the distance. It reflects the difference between the Boosted design and the Standard one. Moreover, it shows the difference between the different Boosted versions implemented 
- Figures 10 and 11: these figures show the results of the proofs of temperature and humidity variation. They allowed to assess the behavior of the sensors under different operating conditions
- Figure 12: it shows another important parameter in textiles, washability

Regarding the methods for the assessment of comfort, conformability, flexibility, and lightness:

Since the sensors initially are designed to be attached into clothing or textiles surfaces such as armchairs, armchairs, curtains or automotive upholstery, we have removed the references to comfort, conformability and lightness from the “Conclusions” section. We have added the following paragraph to the “Results and Discussion” section and we have moved Figure 5 (Figure 9 in the last version) to below the same paragraph:

“As aforementioned, the sensor is initially designed to be attached into clothing or textiles surfaces such as armchairs, curtains or automotive upholstery. In this case, the most important textile feature is flexibility. Several bend radiuses were proved and the response was successful up to radiuses lower than 3 cm. The driver had to be calibrated for every test. Figure 9 shows the flexibility of the resulting 3DBS_Embroidery sensor, assessing the flexibility feature of the e-textile sensor.”   

The word sensibility has been changed to sensitivity. It was a mistake in the abstract, not in the rest of the paper.

- Comment #3:  The whole paper is written like a novel sensor is being presented. The abstract reads “Moreover, three manufacturing techniques have been considered as alternatives: screen printing with conductive ink, embroidery with conductive thread and thermoseal with conductive fabric. The main critical parameters have been analyzed for each prototype including the sensibility of the sensor, which is an important and specific parameter of this type of sensor”, but the response of the authors insists with “this kind of sensors have been validated and the results published in previous publications (referenced in the same paper).” They might refer to reference 22, in which there is no validation of gesture recognition either, but of the electrical and mechanical characteristics of the sensors.  There is a big discrepancy between what the authors understand by validation (of a solution of a hand recognition touchless sensor) and my interpretation of the word.  The comment 2 still holds completely.

Response:

We have extended the section “3. Results and Discussion” including the results of the validations we have carried out regarding gesture recognition. 

First, we characterized the sensors using the artificial hand in a static way. It was located at different positions and heights measuring the signal deviation. The following paragraphs and figures have been added a to the paper:

“For the characterization of the sensors, the following protocol has been established. An artificial hand has been placed at different positions on the X, Y and Z axis with respect to the center of the sensor. For each position, the signal deviation data provided by the device has been registered. For the sake of simplicity, only the measurements taken from the PCB sensor and the screen-printed sensor (3DBS_Screen) are shown in Figure 14. It shows the curves obtained by moving the artificial hand in the X axis in the [-5 cm, 5 cm] interval in 1 cm steps for the following values of Z: 1 cm, 3 cm and 5 cm. In all the cases, the value of Y = 0.

Figure 14. Characterization of the sensors using an artificial hand: a) PCB reference sensor fixing Z = 0 position. b) Textile sensor fixing Z = 0 position. c) PCB reference sensor fixing Z = 3 cm position. d) Textile sensor fixing Z = 3 cm position. e) PCB reference sensor fixing Z = 5 cm position. f) Textile sensor fixing Z = 5 cm position.

As can be seen, the results for the textile sensor are not as smooth as in the PCB sensor due to the textile structure itself. On the other hand, a conductivity variation is observed in the textile sensor, as opposed to the case of the PCB sensor. This behavior is due to the asymmetry of the east and the west sensor. The design of the east sensor tracks is shifted to the left favoring a shorter track for the west sensor. The different tests carried out showed the same behavior. Despite this, the functioning of the gesture recognition is not affected, since the variations are high enough to be detected in the five sensors. The rest of the sensors showed a similar behavior.”

Next, an experiment using different subjects was carried out. They were asked to make certain gestures on the sensor. The detection rate was evaluated after several repetitions. The following text has been added to the paper:

“Next, a validation of the sensors was carried out with different subjects. The objective was to measure the detection rate of several gestures. Figure 15 shows two of the gestures used in the validation procedure. The software used, AUREA, is supplied by Microchip.

Figure 15. Two of the gestures used in the validation: a) flick from north to south and b) flick from west to east.

In the validation procedure, 10 subjects participated in the experiment. Initially, they were completely unaware of the operation of the sensor. They were allowed to interact with the sensor for 10 minutes. Afterwards, they were asked to perform different gestures detectable by the sensor. After a calibration, the subjects were asked to make 10 repetitions of the following movement types: flick from west to east, flick from east to west, flick from north to south, flick from south to north, clockwise circular and counterclockwise circular. The results of the detection rate for the 3DBS_Screen sensor are shown in Table 6.

Table 6. Gesture conducted tests results for the 3DBS_Screen sensor

Gesture S. 0 S. 1 S. 2 S. 3 S. 4 S. 5 S. 6 S. 7 S. 8 S. 9 %
Flick W-E 90% 90% 80% 80% 90% 80% 70% 90% 90% 80% 84,00%
Flick S-N 80% 100% 80% 100% 60% 80% 70% 80% 70% 70% 79,00%
Flick N-S 90% 100% 90% 80% 80% 90% 90% 80% 70% 60% 83,00%
Flick E-W 90% 90% 90% 90% 90% 90% 90% 90% 70% 80% 87,00%
Counterclockwise 60% 70% 70% 70% 70% 60% 60% 70% 60% 60% 65,00%
Clockwise 70% 80% 70% 80% 80% 70% 70% 60% 70% 80% 73,00%

The results show that most of the gestures are detected successfully. All the flick-type gestures were detected with a percentage higher than 75%. In contrast, the clockwise and counterclockwise gestures have a lower detection rate. This is due to the higher complexity of the circular movement. Besides, the counterclockwise is more difficult to detect than the clockwise movement.

Table 7 shows the comparison of the results obtained by the three proposed sensors and the PCB sensor.

Table 7. Comparison between proposed sensors

Gesture 3DBS_Fabric 3DBS_Screen 3DBS_Embroidery PCB
Flick W-E 84% 84,00% 86% 89%
Flick S-N 81% 79,00% 81% 81%
Flick N-S 84% 83,00% 85% 86%
Flick E-W 88% 87,00% 89% 90%
Counterclockwise 59% 65,00% 73% 86%
Clockwise 76% 73,00% 74% 84%

It can be observed that although the three proposed sensors have slightly worst performance than the PCB sensor, most of the gestures were detected successfully. The proposed sensors have better performance with flick gestures rather than with couterclockwise and clockwise gestures. Among the proposed sensors, the 3DBS_Embroidery showed the best performance.”

- Comment #4:  The authors changed my words to “wearable sensing textiles itself cannot be considered a motivation for a scientific study”. I said “The demand of wearable sensing textiles itself cannot be considered a motivation for a scientific study”.  In other words, one can argue that there is a need for seamless gesture-recognition wearable technology, for physiological signals from wearable devices, among other scientific issues, but the demand of wearable sensing textiles (as it is a too broad term)  is hard to understand as a scientific motivation for a research study.

Response:

We are facing a subjective question subject to the personal interpretation of each one. In this paper, we propose a new development, but not thinking of satisfying the demand of a hypothetical market, but thinking of contributing in a future research prototype or directly in a future manufactured product. Smart Textiles and wearable devices based on textiles are right now in an initial development phase and the involved Technology Transfer cannot be immediate. Some questions remain open, such as volume and hardiness, that are totally related with the batteries and electronic components needed to supply energy, process information and send data. 

All these open issues will be solved in the future, reducing sizes and consumptions of the microelectronic components. Therefore, we understand that although currently there is not a firm demand for this type of solution, it is necessary that the scientific community experiment with this type of technologies. 

We have been working in capacitive based textile sensors for 4 years developing textile touchless sensors in AITEX, a Textile Technological Institute. We have tested several textile substrates together with several conductive inks and threads. We have designed a lot of prototypes with a lot of different parameters, some of them working and other not. All this research and experimentation has been carried out following scientific processes with the main objective of ending in manufactured products (Technology Transfer). 

An interesting article dealing with this topic can be found in [3]. It states that “in modern times science is a necessary condition for technology and vice versa”.

3. Moravcsik, M.J. The role of science in technology transfer. Res. Policy 1983, 12, 287–296

- Comment #5:  The manuscript (first sentence of the abstract) starts with a typo “There in an interest”.

Response:

The error has been corrected.

Besides, hand recognition has been changed by gesture recognition.

Yours sincerely,
Eduardo García-Breijo
Full Professor
Universitat Politècnica de València, Spain
